# Revisiting Instruction Fine-tuned Model Evaluation to Guide Industrial Applications

**Manuel Faysse**[1,3]    **Gautier Viaud**[1]    **Céline Hudelot**[3]    **Pierre Colombo**[2,3]

[1]Illuin Technology, Paris, France    [2]Equall, Paris, France

[3]MICS, CentraleSupélec, Université Paris-Saclay, France

manuel.faysse@centralesupelec.fr

## Abstract

Instruction Fine-Tuning (IFT) is a powerful paradigm that strengthens the zero-shot capabilities of Large Language Models (LLMs), but in doing so induces new evaluation metric requirements. We show LLM-based metrics to be well adapted to these requirements, and leverage them to conduct an investigation of task-specialization strategies, quantifying the trade-offs that emerge in practical industrial settings. Our findings offer practitioners actionable insights for real-world IFT model deployment.

## 1 Introduction

Adapting pre-trained language models (LMs) for specific applications is central in industrial NLP to unlock task-specific performance gains and strengthen model alignment with industry requirements. A paradigm gaining traction is the use of instruction fine-tuned (IFT) models, LMs capable of following arbitrary instructions expressed in natural language (Wei et al., 2022a; Sanh et al., 2022; Ouyang et al., 2022).

Researchers primarily concentrate on improving general-purpose IFT models to be used as versatile agents capable of executing instructions expressed in natural language (Li et al., 2023; Zhou et al., 2023; Xu et al., 2023a). In an industrial setting, prompting ChatGPT to improve the wording of an email, or to assist with a code snippet would be instances of this zero-shot utilization scenario, which we define as $\mathcal{S}_0$. Critical industrial LLM applications may however not always align with $\mathcal{S}_0$, and often prioritize two other settings. The first scenario, $\mathcal{S}_1$, requires extending a generalist IFT model's capabilities to new specific tasks not included in the original instruction training set. The second scenario, $\mathcal{S}_2$, centers around converting IFT models into specialized models proficient *exclusively* on specific tasks. In $\mathcal{S}_1$ for instance, a large company may want an LLM assistant for internal employee use, and decide to extend an openly available Chat model by training it to write memos with a specific templating scheme, to respond to internal FAQs, and to use internal coding tools, all the while retaining the original chat assistant's general purpose abilities. In $\mathcal{S}_2$, that same company is only interested in a given specific task; extracting specific information from business documents, and specializes an IFT model for that purpose, aiming to leverage prompting and the generalization capabilities of the model for a more data-efficient training.

In this paper, we thoroughly examine $\mathcal{S}_1$ and $\mathcal{S}_2$ by investigating the learning dynamics of specializing IFT models through a practical lens. To ensure the reliability of our tooling and the rigor of our conclusions, we first undertake a critical assessment of the current evaluation practices employed for IFT models. Formally, our contributions are:

**Contribution 1.** IFT models are designed to handle tasks of diverse natures and varying difficulties. However, current metrics used to measure their performance are often task-specific (Zellers et al., 2019; Gao et al., 2021), or rely on automatic metrics designed for other intended purposes (Papineni et al., 2002; Lin, 2004). To address this limitation, we introduce two new requirements for metrics used to evaluate IFT models: Comparability Across Task (CAT) and Task and Format Agnostism (TFA). CAT imposes for metric scores to exhibit consistency across a diverse set of generative tasks, in contrast to the sole traditional focus of consistency within a specific task. TFA defines the need for metrics to demonstrate robustness to variations in the output formats. By highlighting the shortcomings of existing metrics in meeting CAT and TFA, we present compelling evidence that using LLMs as scoring agents is a viable evaluation alternative of IFT models.

**Contribution 2.** We approach our examination of $\mathcal{S}_1$ and $\mathcal{S}_2$ from a practical perspective and focus on

the trade-off between data availability and overall performance. Our analysis uncovers two distinct phases of learning during IFT model specialization: learning to format, and learning to solve tasks. Subsequently, we showcase how practitioners can (i) leverage synthetic data to facilitate learning the desired formatting aspects and (ii) use IFT models to reduce the need of expert data in industrial scenarios. Our study provides practical insights and actionable recommendations to practitioners looking to deploy IFT models in production settings.[1]

## 2 Re-evaluating IFT Model Evaluation

### 2.1 What Should Good Scorers Measure?

In scenarios $\mathcal{S}_0$, $\mathcal{S}_1$, and $\mathcal{S}_2$, IFT models are trained to perform generative tasks. Unlike models designed for single tasks with known output formats, IFT models have the capacity to generate diverse valid responses across different tasks and formats (Ouyang et al., 2022). The novel capabilities of IFT models impose new considerations when selecting an automatic evaluation metric.

**Comparability across tasks (`CAT`).** Standard evaluation metrics aim to fulfill one key requirement: coherence within each task with respect to human judgment (Specia et al., 2010). However, due to the multi-task nature of IFT models, the scores should also be comparable across different tasks (Colombo et al., 2022a; Himmi et al., 2023). In other words, the scoring scale should be absolute and coherent with human preferences on all tasks. To measure the `CAT` we will mix samples of different tasks and compute the Spearman correlation ($\rho$) of their score with human judgment[2]. This requirement is essential in scenarios $\mathcal{S}_0$ and $\mathcal{S}_1$ to measure model performance across different tasks, and make informed decisions regarding the trade-offs between model variants.

**Task and Format-Agnostism (`TFA`).** Evaluation metrics should be robust to artifacts associated with the output format and to the nature of the evaluated task (Liang et al., 2022). Implementing task-specific scoring metrics is not a scalable solution for generalist IFT models. To measure `TFA`, we compute the relative target task improvement between models prompted in a zero-shot manner

and models that mastered the task format (trained on 1000 target task samples). Comparing each metric's `TFA` to human-reported performance improvements allows to grasp the extent to which mastery of the task format influences the metric performance, independently of intrinsic task performance. In industrial scenarios, this requirement is essential as it ensures minimal bias in the evaluation due to training data formatting artifacts. In practice, many datasets that industrial actors may add to the training instruction set ($\mathcal{S}_1$), or fully train a model on ($\mathcal{S}_2$) have specific response formatting that differs from what a zero-shot model will answer, leading to a potentially large formatting bias.

**Comparability intra-task (`CIT`).** While in no way a novel requirement, it is essential for metrics to measure performance consistently within a given task. We verify this by computing the Spearman $\rho$ correlation coefficient between samples of a specific task and human judgments.

In all industrial scenarios for IFT LLMs, rigorous model evaluation is necessarily linked to evaluation metrics that comply with both `CAT` and `TFA`, as well as the standard `CIT` measures.

### 2.2 Existing Metrics

**Current Evaluation.** Currently, two dominant paradigms emerge for assessing the performance of IFT models: (i) relying on reference-matching scoring metrics such as `ROUGE-L` (Lin, 2004), or normalized log-probabilities of class labels in few-shot classification benchmarks (Hendrycks et al., 2021; Gao et al., 2021; Zellers et al., 2019), and (ii) model ranking frameworks, based on pairwise preference comparisons of response quality judged by humans or LLM evaluators (Chiang et al., 2023; Dubois et al., 2023; Gudibande et al., 2023). Language Model based scoring has been shown to be a promising alternative on specific tasks, such as summarization (Liu et al., 2023; Colombo et al., 2022b) or translation (Kocmi and Federmann, 2023; Xu et al., 2023b). Our work extends these findings to showcase the multi-task scoring capabilities of LLMs with respect to `CAT` and `TFA`.

**LMs as Viable Scoring Mechanisms.** Given the inherently open nature of IFT model generation, we adopt a reference-free setting to ensure unbiased evaluation. We present an input prompt and the corresponding generated response to the LLM[3],

---

[1]Code and evaluation datasets are available on `https://github.com/ManuelFay/IFTEval`.

[2]Common when benchmarking metrics (Bhandari et al., 2020; Colombo et al., 2021; Chhun et al., 2022; Staerman et al., 2021; Fabbri et al., 2021; Colombo et al., 2021), we extend the tool to inter-task settings.

[3]We rely on LMs available through the OpenAI API (*i.e.,* GPT4 and GPT3.5). See Sec. A.3 for details.

prompting it to assign a score on a scale of 0 to 10, subsequently scaling it between 0 and 1 to facilitate comparisons with other evaluation metrics.

**Baseline Metrics.** We assess the fulfillment of both CAT and TFA by comparing the proposed metrics against well-established *reference-based* metrics, including ROUGE[4], BScore (Zhang et al., 2020), and SBERT (Reimers and Gurevych, 2019), as well as a *machine learned* metric, the OpenAssistant Reward Model (RM) (Köpf et al., 2023) trained on human preferences.

## 2.3 Experimental setup

**Training an IFT model.** IFT models are trained by fine-tuning a base model on a large instruction corpus, collected either through human annotations (Ouyang et al., 2022; Köpf et al., 2023) or concatenating task-specific datasets (Sanh et al., 2022; Mishra et al., 2022). In line with recent work (Chiang et al., 2023; Wang et al., 2023; Peng et al., 2023), we leverage synthetic data as the base instruction set in our IFT models (Taori et al., 2023).

**Benchmarking automatic metrics.** To benchmark the metrics, we rely on a combination of synthetic and real data. For *synthetic data*, we use the Alpaca GPT4 dataset (Taori et al., 2023), and tag the data in 13 task categories (see Sec. A.1) (*e.g.*, logic, code, rewrite). For *human data*, we focus on tasks with industrial interests. Specifically, we include Natural Language Inference (Williams et al., 2018; Wang et al., 2019), Question Answering (Rajpurkar et al., 2016), NER (Tjong Kim Sang and De Meulder, 2003), and Sentiment Classification (Socher et al., 2013; Agirre et al., 2013)). To build our metric evaluation dataset, we train and run LLaMA-7B models (Touvron et al., 2023a) on varied data mixtures and target tasks. For rigor, we also report scores on the summarization with human feedback dataset from (Stiennon et al., 2022) (SUM).[5]

## 2.4 Experimental results

To better understand the limitation of existing metrics we conduct both single-task analysis to ensure that metrics are able to score tasks reliably as well as multi-task analysis, which is the standard setting for IFT models. Results are reported in Tab. 1.

**CIT Analysis.** From Tab. 1(left, SUM), we observe that the average correlation with human scores for evaluated summaries are higher for LLM models

than with traditionally used metrics. Intra-task correlations on all other *human data* tasks, averaged in CIT lead to similar conclusions.

**CAT Analysis.** Tab. 1(left) shows that all metrics, *with the exception of the GPT4-based metric*, exhibit weak or no correlation in the context of inter-task consistency. While it is true that existing metrics demonstrate the ability to differentiate between good and bad samples within a single task (CIT), their *performance falls short when confronted with the open setting imposed by IFT models*.

**TFA Analysis.** On non-LLM-based metrics, performance gains reported between zero-shot models, and models trained on 1000 target-task samples (Tab. 1(left), TFA) largely exceed the 12.0 % relative improvement of human scores, and demonstrate how format, once learned, unrealistically boosts reference-based metrics which are heavily impacted by format.

**Metric Similarity Analysis.** Fig. 1 displays metric correlation at the sample level on the synthetic dataset. The results align with (Chen et al., 2022), indicating a moderate to high correlation between BERT-based metrics and ROUGE-L. However, all metrics exhibit a low correlation with GPT4, indicating different response features are scored.

**Zoom on GPT4.** Tab.1(right) shows a strong correlation between the results of GPT4-based metrics and the corresponding LLM task abilities reported in Wei et al. (2022b) (Logic and Coding are non-trivial for LLMs, Writing tasks are relatively easier). However, reference-based metrics such as ROUGE suggest the opposite, as they are biased by the high syntactic overlap between model outputs and reference answers in these categories. The GPT3.5 scorer also highly rates results on the Logical Reasoning tasks, contrarily to GPT4. This is due to its lesser ability to spot logical inconsistencies in the evaluated responses (Bubeck et al., 2023), hinting that evaluator models must be capable at the evaluated task themselves in order to produce meaningful scores.

Our findings highlight the inadequacy of existing metrics in quantifying the performance of IFT models, while emphasizing GPT4 as a promising candidate. This performance gap in evaluation capabilities is primarily explained by GPT4's reduced dependence to reference answers, leading to a more coherent and absolute evaluation scale CAT, and an improved robustness to variations in output formatting TFA. *The GPT4 scorer's powerful capabilities*

---

[4]ROUGE-1 is used here, it is best on one-word long labels

[5]More details in Sec. B.1)

Table 1: *(Left)* $\rho$ **correlation between human scores and metrics** on the summarization task (SUM), on the *human data* tasks individually then averaged in (CIT), and on the concatenated human tasks (CAT) to form inter-task settings. (TFA) denotes relative metric improvement after 1000 target task samples are added to the training set. *(Right)* **Metric scores** averaged per *synthetic data* category

| Scorers | SUM | CAT | CIT | TFA | | Scorers | Logic | Code | Memory | Write |
|---------|-----|-----|-----|-----|---|---------|-------|------|--------|-------|
| ROUGE | 0.28 | 0.22 | 0.57 | +513.9 % | | ROUGE | **0.52** | 0.46 | 0.47 | *0.41* |
| BScore | 0.21 | 0.22 | 0.13 | +49.0 % | | BScore | **0.73** | *0.71* | **0.73** | 0.72 |
| SBERT | 0.25 | 0.29 | 0.43 | +86.3 % | | SBERT | 0.80 | 0.74 | **0.84** | *0.74* |
| RM | 0.20 | 0.28 | 0.29 | -44 % | | RM | **0.49** | 0.43 | *0.28* | 0.33 |
| GPT4 | **0.45** | **0.68** | **0.77** | +2.1 % | | GPT4 | *0.71* | 0.79 | 0.93 | **0.97** |
| GPT-3.5 | 0.42 | -0.19 | 0.48 | +9.5 % | | GPT-3.5 | **0.88** | *0.82* | 0.87 | 0.86 |
| Human | 0.54 | - | - | +12.0 % | | | | | | |

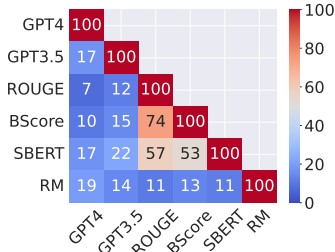

Figure 1: **Spearman $\rho$ between metrics** on synthetic data.

*unlock the study of novel settings traditional metrics would struggle with* (Schaeffer et al., 2023).

## 3 IFT Models for Industrial Applications

### 3.1 $\mathcal{S}_1$: Improving Specific Tasks

In this section, we delve into $\mathcal{S}_1$, which specifically aims to extend an IFT model's capabilities to better perform on specific instructions.

**Setting.** We fine-tune a base 7B-LLM model (Pythia Biderman et al. (2023), Bloom Scao et al. (2022), Falcon Penedo et al. (2023), or LLaMA) using synthetic instructions. In each training iteration, we introduce a selected number of $N$ *real* target task samples into the synthetic dataset. We evaluate the model performance on an independent test subset of the target task.[6]

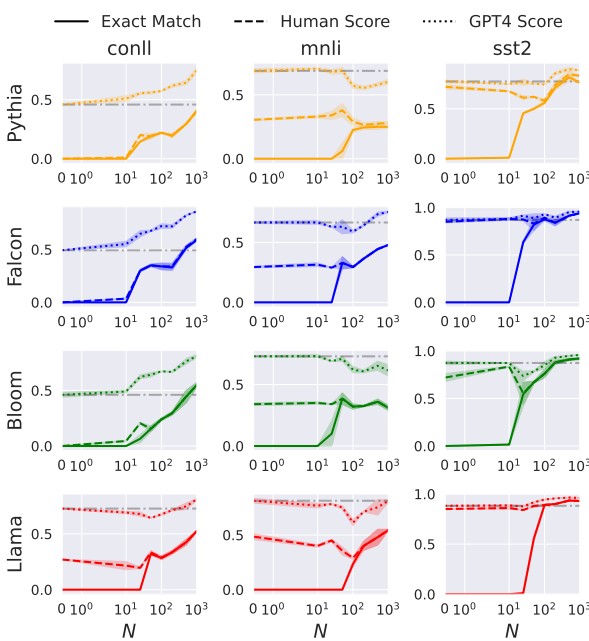

Figure 2: Incorporating $0 \le N \le 1000$ real task samples into IFT model training

---

[6]More experimental details are given in Sec. C.1.1.

**Mastering Format to Foster Understanding.** Fig. 2 shows target task performance as the number of target task samples introduced within the base training set increases. Across all tasks and models, *specialization is biphasic*: first, *task output format* is learned while overall performance remains constant, or even slightly decreases. Only once the format has been mastered, as noted by the spike of the Exact Match, does the model improve upon its *underlying task performance* (Human and GPT4 scores increase), eventually surpassing the original zero-shot performance. It is worth noting that this analysis is made possible by format-agnostic scorers (TFA) that can accurately decouple output format from underlying model performance.

**Measuring Model Forgetting.** Performance on a test split of the Alpaca data shows little to no performance degradation (<1%) caused by the inclusion of new tasks to the training mix (Sec. C.1.2).

**Leveraging Synthetic Data to Learn to Format.** Our findings suggest a straightforward approach to optimizing the use of real examples: *employ synthetic examples to assist the model in mastering the desired format before relying on real samples to enhance overall model performance.* We repeat the previous experiment, replacing the $N$ human-annotated target task training samples ($H$), by GPT4 synthetically-generated samples ($S$), or synthetic samples with random labels ($R$) (Fig. 3) Exact Match shows synthetic or randomly labeled data can indeed be used to learn the desired format, although the better quality human data eventually yields better results with more samples. In ($S+H$), we train on 100 synthetic samples, then on $N$ human-annotated samples. This technique enables the model to master the format before being trained on high-quality data, largely improving human annotated data sample efficiency.

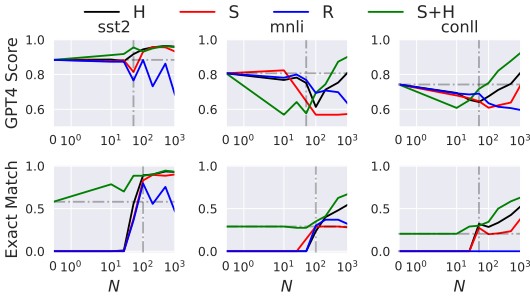

Figure 3: Incorporating $0 \leq N \leq 1000$ (H)uman, (S)ynthetic and (R)andomly labeled synthetic data samples in IFT training set. (S+H) is trained on 100 synthetic samples, then $N$ human data samples.

### 3.2 $\mathcal{S}_2$: IFT models as Task-Specific Solvers

**Setting.** We use four model architectures and, for each architecture, we employ the base model to train an IFT model variant using the synthetic Alpaca dataset. We then fine-tune both the base models and their IFT variants on a subset of $N$ samples drawn from the target task. This setup simulates an industrial scenario in which limited data is available to specialize a model on a unique task, and assesses whether there are benefits to instruction-tuning a base model before fine-tuning it on the target task.

**Results.** Fig. 4 demonstrates that IFT models exhibit enhanced performance in low-data scenarios (when $10 \leq N \leq 200$). Intuitively, IFT models are better able to leverage the task description given in the prompts, thus enabling boosted zero-shot performance (Scao and Rush, 2021). This complements and steers model training when finetuned with small numbers of samples. When more samples are available ($N \geq 200$), the task pattern is sufficiently clear and the benefits of prompting and of the IFT training phase disappear.

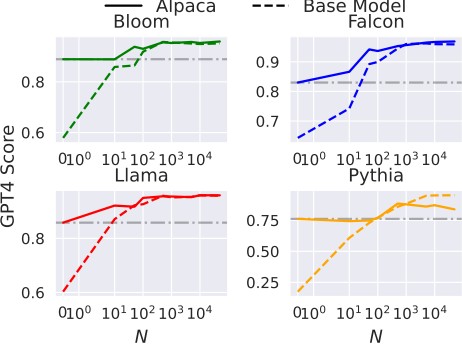

Figure 4: GPT4 score on SST-2 test set after finetuning with $0 \leq N \leq 1000$ samples on a (base) LM or an IFT model. Further experiments can be found in Sec. C.2.2.

This finding aligns with the results presented in Sec. 3.1, emphasizing the potential of synthetic datasets to enhance data efficiency in industrial scenarios.

## 4 Key Takeaways for Practitioners

**Leveraging LLM for evaluation.** Evaluating IFT models is challenging, as it mandates *comparability across tasks* and *format-agnostism*, which standard metrics struggle with. While LLM scoring is not ideal (Limitations in Sec. 4), it is a strong option practitioners should add to their arsenal.

**Leveraging Synthetic Data for Efficient Learning.** LLM-based evaluation uncovers the fact that leveraging synthetic data provides a quick and cost-effective approach to mastering format in low data regimes, with no performance degradation. This methodology proves viable across various scenarios, presenting an opportunity to more efficiently leverage potentially limited amounts of expert annotated data available.

## Limitations

While this paper has argued in favor of using LLM as scorers, important drawbacks remain. The best-performing scorer at the moment, GPT4 is a proprietary, black-box access model, and no guarantees exist that it will remain accessible unchanged over time, leading to reproducibility issues and data privacy concerns. Since model and training data internals are not open-knowledge, analysis of scoring errors and internal model biases is also limited.

Promising openly available alternative models are being developed, either general purpose LLMs aiming to disrupt the hegemony of GPT4 (Touvron et al., 2023b; Bai et al., 2023), or smaller models specialized for automatic evaluation, often attempting to distill GPT4's scoring abilities by training on GPT4 generated scores or scoring explanations (Xu et al., 2023b; Liu et al., 2023). In the latter category, the Prometheus scoring model (Kim et al., 2023), based on Llama2, claims scoring performances on par with GPT4 in terms of human score correlations over a variety of tasks and benchmarks. Eventually, strong Open-Source LLMs should alleviate most of the concerns raised by relying on proprietary black-box models and we hope this work, by shedding light on the importance of LLM scoring, motivates these efforts to build open models with strong scoring abilities.

## Ethics Statement

While this work intends to evaluate scorers across many different tasks and settings, it is essentially English-centric, and no conclusions are drawn about the robustness of LLM scorers in other languages. LLM scoring may also be affected by internal model bias acquired through pretraining or subsequent finetuning, and while efforts are made by OpenAI to mitigate bias, critical applications of LLM evaluation should consider that truly objective evaluation is not attainable.

All data and base models used in this work originate from publicly available sources. The GPT4 Alpaca dataset is a variant of (Taori et al., 2023) built from synthetic data only, collected through the OpenAI API. The non-synthetic data are sourced from manually annotated, widely used datasets for NLP benchmarking. This work does not transgress any usage restrictions associated with these data sources. Base models used are either available through fully open-source licenses (Falcon, Pythia), or licenses with no restrictions for research purposes (LLaMA, Bloom).

We estimate our experiments consumed 5500 GPU V100 hours, using a low-carbon compute cluster, amounting to about 950 kg of $CO_2$ over the course of the project. To reduce the impact to a maximum, all runs are done through the efficient Low-Rank Adaptation training strategy (Hu et al., 2021), and only trained adapter weights are stored to minimize bandwidth and memory usage. API calls to external APIs are cached to minimize redundancies.

## Acknowledgements

This work is partially supported by Illuin Technology, and by a grant from ANRT France. This work was performed using HPC resources from GENCI–IDRIS (Grant 2023-AD011014185). Secondary compute contributions were made on HPC resources (Grant 2023-AD103256) and (Grant 2023-AD101838).

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

## A Ressources

### A.1 Data

The GPT4 Alpaca dataset is collected from `https://huggingface.co/datasets/vicgalle/alpaca-gpt4`.

We define a taxonomy of 13 subtasks that the instructions can fall into: Classify, Code, Answer from Context (Question Answering based on a given passage), Create (Artistically oriented Natural Langiage Generation), Extract, Logic (Reasoning tasks), Answer from Memory (Question Answering from internal model knowledge), Summarize from Memory (Summarization/Explanation from internal model knowledge), Rewrite (Reformulation tasks), Write (Natural Language Generation tasks with a non-artistic goal), Summarize (Summarization tasks given a passage), Translate, and Other. We then tag each of the 52000 instructions with the subtask it falls into using GPT4 as an oracle by prompting it with the classification task. To verify the quality of the tagging, we manually verify a random sample of 100 instructions and find a 93% agreement rate.

The other datasets also originate from the HuggingFace Hub [7][8][9]. We format all tasks as instructions using the following prompts and concatenating the input at the end.

**MNLI** (Williams et al., 2018) Classify the following relationship between the Hypothesis sentence and the Premise sentence, as either Entailment, Contradiction or Neutral.

**QNLI** (Wang et al., 2019; Rajpurkar et al., 2016) Classify whether the given context contains enough information to answer the question (answerable) or not (unanswerable).

**STSB** (Agirre et al., 2013) Give an integer score between 1 and 5, describing how similar sentence1 and sentence2 are. 5 means they are very similar, 1 means they are nothing alike.

**SST2** (Socher et al., 2013) Classify the following sentence as negative or positive.

**CONLL** (Tjong Kim Sang and De Meulder, 2003) Extract locations, persons, and organizations from the text. The output should be formatted as a JSON object with three keys: LOC (locations),

PER (persons), and ORG (organizations). Each key should have a value that is a list of strings. If the text contains no entities of a given type, the corresponding list should be empty.

**SQUADV2** (Rajpurkar et al., 2016) Answer the question depending on the context. You must only answer with one excerpt from the text.

**XSUM** (Narayan et al., 2018) Summarize the following article in a few words.

Finally we split each data task category into 3 sets: a training set, a validation set, and a test set, with 80%, 10%, and 10% of the data respectively.

### A.2 Models

Models are publicly available on the HuggingFace Hub, and are fairly similar decoder GPT architectures. We select the 7 billion parameter version to compare models with similar scales: **LLaMA** (Touvron et al., 2023a), **Falcon** (Penedo et al., 2023), **Pythia** (Biderman et al., 2023), **Bloom** (Scao et al., 2022)).

### A.3 Metrics

Model performance is measured using two family of methods: *reference-based metrics* and *LM metrics*.

**Reference-Based metrics** are the common metrics used in the litterature to evaluate the performance of a generative model on a given task, and offer a measure of the distance between a generation and a reference output. In this category are included the **Exact Match** and F-measures, but also more task-specific metrics based on co-occurence of words betwwen the output and the reference, such as **ROUGE** scores (Lin, 2004) for summarization, BLEU scores for translation (Papineni et al., 2002). To go beyond word matching heuristics, we also baseline neural network scorers. **SentenceBert** models (Reimers and Gurevych, 2019) enables to calculate the cosine similarity between sentence embeddings; in this work a general purpose embedding model (all-mini-lm-l6-v2) is used. **BertScore** computes a F1 score leveraging pairwise cosine similarity between tokens, using a strong encoder model finetuned on an NLI task as shown to be best, here we use `microsoft/deberta-base-mnli` from the HuggingFace Hub (He et al., 2023).

---

[7]`https://huggingface.co/datasets/glue`
[8]`https://huggingface.co/datasets/squad_v2`
[9]`https://huggingface.co/datasets/conll2003`

**LM metrics** are metrics that are computed by directly scoring the output using a language model. In this work, we experiment with **GPT4** and **GPT3.5** as scorers, by providing the evaluated model output and tasking the scorer to grade the quality of the output fom 0 to 10, based on relevance, fluency, factuality, coherence. The scoring prompts are task agnostic and allow to compare the performance of a model on different tasks, or on open-ended tasks where there is no ground truth and traditional literature metrics cannot be used. They also have the advantage of being continuous, which allows to study transitive regime without the "emergence" effect (Schaeffer et al., 2023). Finally, we baseline a **Reward Model (RM)** (Köpf et al., 2023), a model trained on outputting a score designed to reflect a human's appreciation of a judgement. While this can be used as a reference-free scorer, we find increased robustness when reporting the score softmaxed with the RM score of the reference.

**GPT Scoring prompt** To obtain GPT4 and GPT3.5 scores for model responses to a given instruction, we prompt the OpenAI API as such:

> You are a helpful assistant that helps evaluate the quality of two responses to a prompt.
>
> Answer by awarding a score between 0 and 10 to each response, where 0 means the response is completely inappropriate and 10 means the response is very good. A response that is acceptable should never be awarded less than 6 out of 10.
>
> Answer base on the following criteria:
> 1. Is the response grammatically correct?
> 2. Is the response semantically correct?
> 3. Is the response coherent?
> 4. Is the response relevant to the prompt?
>
> Output format (csv):
> <score1 from 0 to 10>,<score2 from 0 to 10>
>
> Rate the responses to the following instruction.
> {prompt}
>
> Response 1: {response1}
>
> Response 2: {response2}

Output:

Response 1 corresponds to the model prediction, and reference 2 to the "gold" label. The scores obtained are then checked for conformity (correct output template, scores between 0 and 10). Finally, they are scaled between 0 and 1.

## A.4 Framework

Code is written using PyTorch and the Transformers library, as well as the PEFT library to train models using low-rank adaptation. Training runs are done on compute clusters with NVIDIA V100 32GB GPUs.

## A.5 Default Model Training

We train models across a wide range of models, training data source and dataset sizes. To stay consistent between runs, we train models for 400 steps with a 128 batch size, achieved through gradient accumulation, a 5e-4 learning rate with linear decay, a warmup of 100 steps, or one epoch (whichever is lowest). We frequently log validation split CE loss and use early stopping to prevent overfitting. Code is fully released at `https://github.com/ManuelFay/IFTEval`. Training runs to completion take about 5 hours on a V100 GPU and 2h on a A100 GPU. This default setup is used to train all models in this work.

The only exception are models trained on datasets with less than 128 samples (less than the batch size). In these cases, we select a batch size of 8.

## B Re-evaluating evaluation

### B.1 Experimental setup

**Summarization dataset** In this dataset (Stiennon et al., 2022), summaries are already generated and associated with human scores. We select 4 summarization models from the dataset, and 200 inputs the 4 models have been evaluated in common on, and that have at least 2 different human annotators. This enables computing Spearman Rank Correlation both between metrics and the mean human annotation, but also between human annotators by averaging the pairwise human correlation scores over all pairs of annotators.

**Synthetic Alpaca dataset** Models are trained in the default conditions explicited in A.5. To construct Spearman rank correlations between evaluators, we train a suite of over 52 models in which the

training data mixture slightly differs (exclusion or partial exclusion of a target category). To do so, we (i) select one of the 13 task categories identified in A.1. (ii) We then build a base training data mixtures by concatenating the training splits from all other categories, respectively the validation splits. (iii) Finally, we randomly sample respectively 0, 10, 100 and 1000 samples from the held-out category's training split and add them to the base mixture. (v) We train one model per data mixture according to the training guidelines in A.5. (v) We evaluate performance with all metrics on the held-out task test split. (vi) The process is repeated for all categories.

**Human-annotated datasets** Models are trained in the default conditions explicited in A.5. To evaluate Spearman correlation between rankers (CAT and CIT scenarios), we train a suite of over 300 models in which the training data mixture slightly differs (exclusion or partial exclusion of a target category). To do so, we (i) select all 13 task categories identified in A.1. (ii) We then build a base training data mixtures by concatenating the training splits from all categories, respectively the validation splits. (iii) We then select a category from the benchmark tasks (MNLI, QNLI, STSB, SST2, Squad, XSum, CONLL). (iv) We randomly sample respectively 0, 10, 25, 50, 100, 200, 500 and 1000 samples from the selected category's training split and add them to the base mixture, and repeat the sampling process 4 times;, to obtain 7*8*4 = 504 data mixtures. (iv) We train one model per data mixture according to the training guidelines in A.5. (v) We evaluate performance with GPT4 on the target task test split. (vi) To compute the Spearman correlations, we aim to reduce dataset specific formatting artefacts and thus only select models trained with less than 100 target task samples in the training mixture. Correlations are computed two ways, either by computing Spearman correlations per task category and averaging the correlation values with human judgement (CIT), or by computing correlations globally without considering task category information (CAT) enabling the study of cross-task scoring comparability.

**TFA** Finally, to compute the TFA, that is the relative performance improvements after format is learned, we select 3 target tasks, MNLI, QNLI, SST2. These datasets are the three datasets we have perfect sample-wise binary human evaluation for from the set of *human data* benchmarks. From the suite of trained models, we select 5 base mod-

els finetuned on the Alpaca dataset, and for each task, 5 models trained on both Alpaca and 1000 samples of the target task. Previous experiments (Fig. 2) have shown that on these datasets, format is generally learned after a few hundred samples. This enables us to compare models that learned target task output format with models that have not, using the various metrics at our disposal. We compute the relative performance improvement of all metrics, and average them in Tab. 1 (TSA). While human evaluation shows model performance has progressed with the introduction of target task samples (12%), non-LLM metrics largely overestimate the boost in response performance, because of their strong bias towards outputs matching the "gold" reference. In low-shot settings when no formatting is learned, it is particularly interesting to use format agnostic scorers like GPT4 to truly analyze a model's comprehension of a task.

## B.2 Obtaining Human Scores on the benchmark data

To facilitate the collection of human annotated model outputs on the non-synthetic data, we simplify the problem by reducing it to a binary choice (Correct / Incorrect) on classification-like datasets (SST2, MNLI, QNLI). We manually observe outputs for each trained model and output task category, and craft pre-tagging heuristics to make the annotation process quicker. These heuristics are empirically built with knowledge of model outputs; for example on zero-shot sentiment classification tasks, Llama models will often answer *"The sentence is Positive"*, instead of *"positive"*, but trivial heuristic functions can assist in tagging these as correct answers. This is facilitated by the fact output structure for a given model/task combination is often very similar (especially at low decoding temperatures), and we can iterate on these heuristic functions until no error is detected at all. For non-classification tasks, we adopt a strict rating scheme as well. NER responses through the CONLL 2003 task are considered correct only when the dictionary contains all correct key/value pairs with respect to the ground truth, but disregarding formatting artefacts. For SquadV2, an extractive task, we report the F1 intersection between the predicted answer span, and the correct answer, similarly as what is done in (Rajpurkar et al., 2016). Finally, for summarization, we report ROUGE scores for XSUM as a reference point, but do not include

these results with the other tasks during correlation computations, and rather compute human correlations using the human annotation dataset from (Stiennon et al., 2022), for which we report results on Table 1 (SUM).

Code is released at https://github.com/ManuelFay/IFTEval.

### B.3 Results on Alpaca test set

The full results per category are reported in Table 2.

### B.4 Automatic Error Analysis

To go further in the analysis and uncover novel insights at large scale, LLM models can for example enable automatic error analysis. We select a few low scoring outputs and prompt the scoring models to detect mistake patterns, or give potential explanations for the generative model's shortcomings On the sentiment classification task, automatic error analysis correctly detects that most errors are false positives, triggered by sarcasm, or "the presence of words that might be associated with positive sentiment in other contexts". For NLI tasks, it detects the tested model has troubles with with "understanding negations, contradictions, and subtle differences in meaning between the premise and hypothesis".

**Sentiment Analysis**  The following is a series of predictions a sentiment classification ML model got wrong. Can you spot any patterns in the types of mistakes the model made and explain in what areas the model needs to improve?

> Sentence: the château would have been benefited from a sharper , cleaner script before it went in front of the camera . Model prediction: The sentence is a positive statement. Ground truth: negative
>
> Sentence: but there 's plenty to offend everyone ... Model prediction: The sentence is positive. Ground truth: negative
>
> Sentence: outtakes in which most of the characters forget their lines and just utter ' uhhh , ' which is better than most of the writing in the movie Model prediction: The sentence is positive. Ground truth: negative
>
> Sentence: the filmmakers ' paws , sad to say , Model prediction: The sentence is positive. Ground truth: negative

> Sentence: in the media Model prediction: The sentence "in the media" is a positive statement. Ground truth: negative
>
> Sentence: profanities Model prediction: The sentence "profanities" is a positive statement. Ground truth: negative

'The model seems to struggle with understanding the context and sarcasm in the sentences. It often misinterprets negative statements as positive, possibly due to the presence of words that might be associated with positive sentiment in other contexts. The model needs to improve in understanding the overall context of the sentences and detecting sarcasm or irony. Additionally, it seems to struggle with shorter sentences or single-word inputs, where it might not have enough information to accurately determine the sentiment.'

**NLI**  The following is a series of predictions a NLI ML model got wrong. Can you spot any patterns in the types of mistakes the model made and explain in what areas the model needs to improve?

> Premise: He pulled his cloak tighter and wished for a moment that he had not shaved his head. Hypothesis: The man pulled his super hero cape around himself to show off. Model prediction: Entailment. Ground truth: neutral
>
> Premise: because i always had to do it and so i just pay someone else to do it and they do the they do the cutting they fertilize they um edge and um i think this year i'm going to have some landscaping put in Hypothesis: I still do all the gardening and landscaping myself. Model prediction: Entailment. Ground truth: contradiction
>
> Premise: He's chosen Meg Ryan. Hypothesis: Jon Doe was chosen. Model prediction: Entailment. Ground truth: contradiction
>
> Premise: yep that's what he's worried about the trees or a bush because lilac bushes they they grow fast some people uh would really like to have them and then the people that do have them they spread and they sprout all over their their lawn Hypothesis: He's not worried about the trees. Lilac bushes take a long time

|                         | GPT4 | GPT3.5 | ROUGE-1 | BERTScore | SBERT | RM    | Soft RM |
|-------------------------|------|--------|---------|-----------|-------|-------|---------|
| Write                   | 0.97 | 0.86   | 0.41    | 0.72      | 0.74  | 3.21  | 0.33    |
| Answer from Context     | 0.94 | 0.86   | 0.55    | 0.76      | 0.80  | 1.29  | 0.45    |
| Answer from Memory      | 0.93 | 0.87   | 0.47    | 0.73      | 0.84  | 2.78  | 0.28    |
| Extract                 | 0.92 | 0.88   | 0.54    | 0.77      | 0.83  | 1.61  | 0.52    |
| Summarize from Memory   | 0.92 | 0.85   | 0.45    | 0.73      | 0.84  | 3.21  | 0.43    |
| Summarize from Context  | 0.89 | 0.86   | 0.54    | 0.79      | 0.84  | 1.47  | 0.42    |
| Rewrite                 | 0.89 | 0.87   | 0.68    | 0.86      | 0.82  | -0.20 | 0.47    |
| Translate               | 0.88 | 0.94   | 0.58    | 0.81      | 0.71  | 1.78  | 0.44    |
| Create                  | 0.85 | 0.78   | 0.34    | 0.67      | 0.62  | -0.20 | 0.27    |
| Classify                | 0.85 | 0.76   | 0.50    | 0.73      | 0.72  | 0.74  | 0.43    |
| Code                    | 0.79 | 0.82   | 0.46    | 0.71      | 0.74  | 0.73  | 0.43    |
| Logic Reasoning         | 0.71 | 0.88   | 0.52    | 0.73      | 0.80  | 0.33  | 0.49    |

Table 2: Averaged score over samples and models of each metric on each category. Category *Other* is not listed as it consists in synthetic instructions often not answerable by LLMs without external tooling.

to grow. Model prediction: Entailment. Ground truth: contradiction

Premise: Trying Your Luck Hypothesis: This is not a game of luck, but one of pure skill. Model prediction: Entailment. Ground truth: contradiction

Premise: He bent down to study the tiny little jeweled gears. Hypothesis: The were no jewels on the gears he examined. Model prediction: Entailment. Ground truth: contradiction

Premise: As a basic guide, the symbols below have been used to indicate high-season rates in Hong Kong dollars, based on double occupancy, with bath or shower. Hypothesis: This page intentionally left blank. Model prediction: Entailment. Ground truth: contradiction

Premise: In fact, European nations need to do some serious fiscal housecleaning. Hypothesis: There needs to be some serious fiscal housecleaning by European nations. Model prediction: The relationship between the Premise sentence and the Hypothesis sentence is Neutral. The two sentences are not contradictory or entailing. Ground truth: entailment

Premise: It means that they gather and interpret their material fairly and argue about its interpretations rationally. Hypothesis: They gather and interpret the material fairly and argue rationally about

it. Model prediction: The relationship between the Premise sentence and the Hypothesis sentence is Neutral. Both sentences are independent of each other and do not entail or contradict each other. Ground truth: entailment

Premise: well that's pretty typical though uh i don't uh i don't guess it's going to be any much different uh than than it has been in the past so i expect uh July and August we'll see our or uh share of hundred degree days Hypothesis: We don't ever see the temperature get into the hundreds. Model prediction: Entailment. Ground truth: contradiction

Premise: They would burn to the ground by morning. Hypothesis: They would not burn to the ground by morning. Model prediction: Entailment. Ground truth: contradiction

Premise: A rusty iron gate swinging dismally on its hinges! Hypothesis: The iron gate did not move at all. Model prediction: Entailment. Ground truth: contradiction

The model seems to have a pattern of incorrectly predicting entailment when the ground truth is either neutral or contradiction. It appears to struggle with understanding negations, contradictions, and subtle differences in meaning between the premise and hypothesis. The model also seems to have difficulty with understanding the context and re-

lationships between entities mentioned in the sentences. To improve, the model needs to focus on better understanding negations, identifying contradictions, and recognizing subtle differences in meaning and context between sentences. Additionally, it should work on better identifying the relationships between entities and the overall context of the sentences.

### B.5 Correlations on the non-synthetic human tasks

Table 3 and Table 4 display the scorer correlations between themselves on the non-synthetic datasets in both the `CAT` setting (Table 3), and the `CIT` setting (Table 4). Results are consistent with Fig. 1, with BertScore, SentenceBert, Rouge all displaying greater correlations between themselves than with reference-free metrics, and GPT4 standing out both on scoring within each tasks (CIT) and between tasks (CAT).

## C IFT Models For Industrial Applications

### C.1 $S_1$: Improving Specific Tasks

#### C.1.1 Experimental setup

Models are trained in the default conditions explicited in Sec. A.5. To evaluate target task performance as target task samples are progressively introduced in the generalist synthetic training set, we train a suite, we (i) select all 13 task categories identified in Sec. A.1. (ii) We then build a base training data mixtures by concatenating the training splits from all categories, respectively the validation splits. (iii) We then select a category from the benchmark tasks (MNLI, QNLI, STSB, SST2, SquadV2, Xsum, CONLL). (iv) We randomly sample respectively 0, 10, 25, 50, 100, 200, 500 and 1000 samples from the selected category's training split and add them to the base mixture, and repeat the sampling process 4 times;, to obtain 7*8*4 = 504 data mixtures. (iv) We train one model per data mixture according to the training guidelines in Sec. A.5. (v) We evaluate performance with all metrics listed in Sec. A.3 on the target task test split.

#### C.1.2 Performance degradation

Full performance degradation results after 1000 samples of task-specific data are integrated within the synthetic test set are shown in Tab. 5.

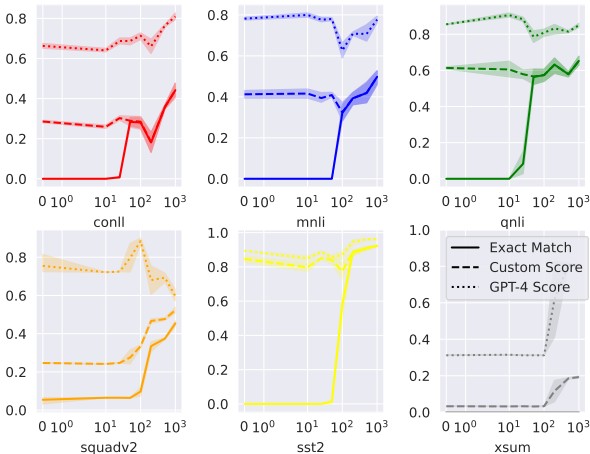

Figure 5: LLaMA performance after incorporating $0 \leq N \leq 1000$ real task samples from respectively CONLL, MNLI, QNLI, SquadV2, SST2 and XSUM into the base training mixture

### C.2 $S_2$: IFT models as Task-Specific Solvers

#### C.2.1 Experimental setup

Models are trained in the default conditions explicited in A.5. Starting from either a base model, or an instruction fine-tuned model, we evaluate target task performance as the model is fine-tuned solely on a varying number of target task training samples. We (i) select a target task from the real data benchmark tasks. (ii) We then sample a set of $N \in \{0, 10, 25, 50, 100, 200, 500, 1000\}$ samples from the training set, and pick 100 samples to serve as a validation split. (iii) We repeat the sampling process 4 times to mitigate variations between runs. (iv) We train one model per data mixture according to the training guidelines in A.5. (v) We evaluate performance with all metrics listed in A.3 on the target task test split.

#### C.2.2 Extra models

To evaluate the impact of IFT quality of the findings of the $S_2$ experiment, we repeat the experiment using an "off-the-shelf" strong IFT model from the HuggingFace Hub. Notably, we select the 7B variant of (Chiang et al., 2023) for the Llama variant, BloomZ for Bloom (Muennighoff et al., 2022), Falcon-Instruct for Falcon (Penedo et al., 2023) and an IFT version of Pythia trained by (Köpf et al., 2023). Results reported in Figure 6 show similar dynamics between the off-the-shelf models and the variants Instruction Fine-tuned in this work for one epoch on the Alpaca dataset (Taori et al., 2023).

|        | ROUGE-1 | GPT4  | GPT3.5 | BScore | SBert | RM    | Human |
|--------|---------|-------|--------|--------|-------|-------|-------|
| ROUGE-1 | 1.00    | -0.16 | 0.46   | 0.73   | 0.85  | -0.77 | 0.22  |
| GPT4    | -0.16   | 1.00  | -0.28  | 0.10   | -0.04 | 0.56  | 0.68  |
| GPT3.5  | 0.46    | -0.28 | 1.00   | -0.00  | 0.16  | -0.37 | -0.19 |
| BScore  | 0.73    | 0.10  | -0.00  | 1.00   | 0.91  | -0.61 | 0.22  |
| SBert   | 0.85    | -0.04 | 0.16   | 0.91   | 1.00  | -0.69 | 0.29  |
| RM      | -0.77   | 0.56  | -0.37  | -0.61  | -0.69 | 1.00  | 0.28  |
| Human   | 0.22    | 0.68  | -0.19  | 0.22   | 0.29  | 0.28  | 1.00  |

Table 3: **Spearman $\rho$ between scorers** on non-synthetic data in the CAT setting.

|        | ROUGE-1 | GPT4  | GPT3.5 | BScore | SBert | RM    | Human |
|--------|---------|-------|--------|--------|-------|-------|-------|
| ROUGE-1 | 1.00    | 0.50  | 0.33   | 0.63   | 0.90  | -0.10 | 0.57  |
| GPT4    | 0.50    | 1.00  | 0.66   | 0.07   | 0.35  | 0.40  | 0.77  |
| GPT3.5  | 0.33    | 0.66  | 1.00   | -0.10  | 0.16  | 0.24  | 0.48  |
| BScore  | 0.63    | 0.07  | -0.10  | 1.00   | 0.66  | -0.43 | 0.13  |
| SBert   | 0.90    | 0.35  | 0.16   | 0.66   | 1.00  | -0.18 | 0.43  |
| RM      | -0.10   | 0.40  | 0.24   | -0.43  | -0.18 | 1.00  | 0.29  |
| Human   | 0.57    | 0.77  | 0.48   | 0.13   | 0.43  | 0.29  | 1.00  |

Table 4: **Spearman $\rho$ between metrics** on non-synthetic data in the CIT setting.

|        | Base | sst2 | conll | mnli | Average |
|--------|------|------|-------|------|---------|
| Bloom  | 0.75 | 0.74 | 0.74  | 0.73 | 0.73    |
| Falcon | 0.87 | 0.86 | 0.87  | 0.84 | 0.85    |
| Llama  | 0.84 | 0.85 | 0.86  | 0.86 | 0.86    |
| Pythia | 0.67 | 0.66 | 0.66  | 0.66 | 0.66    |

Table 5: Performance degradation on the Alpaca test set, after introducing 1000 samples of specialized data to the train set. Base denotes the reference scores, computed from the initial zero-shot performance.

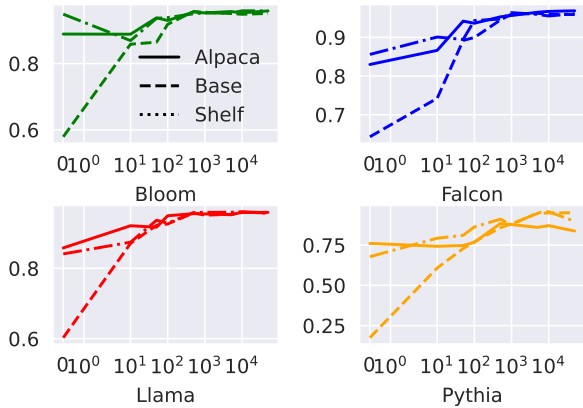

Figure 6: GPT4 score on SST-2 test set after finetuning with $0 \leq N \leq 1000$ samples on a (base) LM, an "off-the-shelf" IFT model, and an IFT model fine-tuned on Alpaca.