# OpenReview forum: "Revisiting Instruction Fine-tuned Model Evaluation to Guide Industrial Applications"
_EMNLP/2023/Conference — EMNLP 2023 Main_

### Official Review · Reviewer_52Xn · 2023-08-04

**Typos Grammar Style And Presentation Improvements:** The examples in the appendix (such as…
**Soundness:** 3

**Excitement:**

4: Strong: This paper deepens the understanding of some phenomenon or lowers the barriers to an existing research direction.

**Paper Topic And Main Contributions:**

This paper discusses the evaluation methods and application scenarios of Instruction Fine-tuning models (IFT models) in industrial applications. The paper proposes two new evaluation metrics, CAT and TFA, and verifies the feasibility of these metrics using LLMs as scoring agents. CAT requires that the evaluation metrics be consistent across different tasks, not just within a specific task. TFA requires that the evaluation metrics be robust to changes in the output format. The authors found that using LLMs as scoring agents is a viable alternative method for evaluating IFT models. In addition, the paper presents a method for improving model learning efficiency using synthetic data when data is limited.

**Questions For The Authors:**

How will the model update and iterative development affect the fairness of the model's scoring? For example, when the scores of the old and new versions are inconsistent.

**Reasons To Accept:**

1. Two new evaluation metrics and a comprehensive method for evaluating the performance of IFT models. In addition, a large number of experiments were conducted to verify the effectiveness of the metrics.
2. Strong evaluation results with detailed experimental setup to reproduce the results.
3. The paper proposes using synthetic data to learn the required format information and improve the performance of IFT models in low-data environments, which is reasonable.


**Reasons To Reject:**

Some experimental results are not well explained and lack justification. For example, C.2.

**Reproducibility:**

4: Could mostly reproduce the results, but there may be some variation because of sample variance or minor variations in their interpretation of the protocol or method.

**Reviewer Confidence:**

3: Pretty sure, but there's a chance I missed something. Although I have a good feel for this area in general, I did not carefully check the paper's details, e.g., the math, experimental design, or novelty.

---

> ### Author Rebuttal · Authors · 2023-08-28
>
> We thank the reviewer for the extensive review and in-depth look at our work. The paper contributions are found to be interesting and clearly stated, and the review highlights our strong experimental results, as well as our findings on data-efficient training strategies.
>
> ### Typos Grammar Style And Presentation Improvements
>
> We thank the reviewer for taking the time to look through the supplementary material, and will make sure to update it to make it more readable (notably examples in B4).
>
> ### Need for experimental clarifications (appendix C.2)
>
> The Appendix C.2 experiment, referred to in main paper Section 3.2  assesses whether fine-tuning an instruction fine-tuned model for a target task, rather than starting from a base model, helps in terms of data efficiency. We find it is the case in low data regimes (typically with less than 200 target task samples, l279), and this result holds across various model architectures. With an extra page in the main paper, we will be able to move experimental details from the appendix to clarify the motivations and setup of this experiment. We will also justify why it is the case; namely that prompting instruction following models unlocks increased zero-shot performance by leveraging the task description to assist the model’s pattern understanding capabilities (Scao, 2021).
>
> Additionally, we will make sure to move some details from the appendix and expand on and justify the rest of the experimental results in both Section 1 and 2 within the newly available page.
>
> ### Interrogations on the consistency of LLM-based rating between model versions
>
> We acknowledge the point raised by the reviewer in the “Limitations” section of our work. Along with other drawbacks, no guarantees are given that black box proprietary models like GPT-4 will remain available and stable over time and this is a real drawback if we want to standardize LLM evaluation across time. OpenAI is aware of the issues caused by inconsistent behavior of its models over time and offers guaranteed long term support for some model checkpoints, which may help, but more realistically, the solution should come from the release of capable open-source alternatives, which is a trend we see with the Llama or Falcon model families for instance.
>
> To expand on this point, we ran extra evaluation experiments using the open source “Llama-2-70B-Chat” model as an evaluator, released after the initial submission date. Our initial results demonstrate this model is capable of outputting evaluation scores that fulfill our CAT and TFA requirements, offering an alternative to reference-based metrics without most of the drawbacks related to using a black-box model (see “Limitations” l304, l312). These interesting results will be expanded and added to the final manuscript.
>
> We thank Reviewer 52xn once again for the comments and suggestions and hope our clarifications reinforce the soundness of our work.

---

### Official Review · Reviewer_NrxC · 2023-08-05

**Soundness:** 3

**Excitement:**

4: Strong: This paper deepens the understanding of some phenomenon or lowers the barriers to an existing research direction.

**Paper Topic And Main Contributions:**

The paper investigates the evaluation metrics of instruction-tuned models. It first presents two new requirements for industrial deployment settings of instruction-tuned models: comparability across task and task format agnostism. Through experimental evaluation of instruction-tuned models, the authors compare LLM-based scorer metrics with conventional automatic metrics (like ROUGE, BERT-Score, etc) and find LLM-based scorers like GPT4 have better correlations with human scores. The paper advocates for LLM-based metric evaluation for instruction-tuned models. For industrial applications using instruction-tuned models, the paper also presents two scenarios: improving specific tasks and using instruction models as task-specific solvers.


The main contribution comes from examining the specific evaluation requirements of instruction-tuned models and empirically showing conventional evaluation metrics do not meet the needs and instead one should use LLM-based scorers.

**Reasons To Accept:**

The paper provides two interesting angles to examine the evaluation of instruction-tuned models. It empirically show the LLM-based metrics can meet the new requirements and would suit industrial applications.

**Reasons To Reject:**

More ablation on why conventional metrics are inferior to GPT4 is preferred, especially for the proposed requirements analysis (CIT, CAT, TFA). Current evaluation only presents results instead of explaining them.

**Reproducibility:**

4: Could mostly reproduce the results, but there may be some variation because of sample variance or minor variations in their interpretation of the protocol or method.

**Reviewer Confidence:**

2: Willing to defend my evaluation, but it is fairly likely that I missed some details, didn't understand some central points, or can't be sure about the novelty of the work.

---

> ### Author Rebuttal · Authors · 2023-08-28
>
> We thank the reviewer for the extensive reading of our work and constructive review. We are glad that reviewer NrxC  finds the concepts behind this research project to be of interest; providing new angles to examine the evaluation of instruction-tuned models and empirically showing the LLM-based metrics can meet the new requirements and would benefit industrial applications.
>
>
> ### On the necessity for more detailed explanations of Section 1 results
>
>
> Currently, insights, notably about GPT4 superiority, are briefly given in Section 2.4 and more details are present in the supplementary material (B.1, l839). We explain metrics based on a distance metric to a gold reference are heavily biased by the syntactic overlap between the prediction and the reference (Zoom on GPT4 subsection, B.1). On tasks where a good response may largely differ from the gold reference (eg. creative writing), the response is penalized by reference-based scorers, and the inverse also holds true (Table 1, Table 2). Additionally, response format heavily biases reference-based metrics (TFA analysis, l194). We show LLM based metrics rely on different response features (“Metric Similarity Analysis” subsection l210, and appendix B.5) and are less sensitive to the gold answer and the output format. Extra results are also given in the Appendix, typically metric explanations (A.3), detailed results per evaluation category (Table 2, B.3), and insights about the criteria (TFA in B.1, l839).  It is thus for their capacity to be less sensitive to bias-introduced by the reference, that LLM metrics are more robust to format (TFA) and coherent between tasks (CAT). We will further detail this point in the main paper upon acceptance.
>
>
> Another insight we did not expand on in the paper by lack of space, that also explains the dominance of GPT-4, can be seen when observing GPT3.5 logic reasoning scores which are very good when compared to the scores GPT-4 attributed. This is simply due to the fact GPT3.5 is not as capable as GPT-4 on such tasks, leading to it not being able to spot logical inconsistencies in the model response as well as GPT-4 does. This hints that evaluator models must be capable at the evaluated task themselves in order to accurately score other models. We will expand on this point in appendix B.3, supported by evidence from Table 1 (main paper) and Table 2 (appendix).
>
>
> We thank the reviewer for highlighting the need for clarifications we will add to the final manuscript, and hope these explanations will make for a more coherent and understandable work.

---

### Official Review · Reviewer_9tn7 · 2023-08-07

**Soundness:** 3

**Excitement:**

3: Ambivalent: It has merits (e.g., it reports state-of-the-art results, the idea is nice), but there are key weaknesses (e.g., it describes incremental work), and it can significantly benefit from another round of revision. However, I won't object to accepting it if my co-reviewers champion it.

**Paper Topic And Main Contributions:**

The paper proposes two novel measures to investigate instruction-following LLMs. The idea is to be able to investigate performance across different instruction following tasks in a coherent way.

**Reasons To Accept:**

- The paper proposes novel measures to investigate performances of instruction-following LLMs .
- Pilot experiments are solid and show that the method is applicable.

**Reasons To Reject:**

- The paper fails to link how the novel measures should be linked to scenarios that are described in the introduction: S_0, S_1, and S_2.
- The paper suggests that there is a need to repurpose an Instruction-following LLM to a specific task. The reason is not sufficiently explained. Saying that this is an industrial need is not sufficient without examples.

**Reproducibility:**

4: Could mostly reproduce the results, but there may be some variation because of sample variance or minor variations in their interpretation of the protocol or method.

**Reviewer Confidence:**

4: Quite sure. I tried to check the important points carefully. It's unlikely, though conceivable, that I missed something that should affect my ratings.

---

> ### Author Rebuttal · Authors · 2023-08-28
>
> We thank Reviewer 9tn7 for their time spent on reviewing our paper. We are glad that the reviewer acknowledges the novelty of our introduced evaluation constraints and our efforts in designing solid, replicable experiments. We answer below the concerns of the reviewer:
>
> ### On the connection between the evaluation requirements and the identified scenario.
>
>
> As stated in paragraph 1 of our introduction, our paper studies LLM performance in various realistic industrial training scenarios, and in order to draw rigorous conclusions, reexamines how to accurately evaluate them by imposing novel constraints.
>
> Comparability Across Task (CAT) is essential for evaluating IFT models in scenarios S_0 and S_1. Given the multi-task nature of IFT models, it is crucial for evaluation measures to score using coherent evaluation scales across tasks, in order to enable comparing model performance as a whole. This necessity is introduced in paragraph l82, and  illustrated in Table 1  (right) and l202 (as well as Appendix Table 2).  Typically, Table 1 shows GPT-4 evaluation estimates the evaluated models are better at redaction tasks (“Write”), than at logical reasoning tasks (“Logic”).  Furthermore, performance trade-offs occur and training to improve performance on a specific task may lead to degradation on other tasks (l247, detailed in Supplementary Material C.1.2). For instance, we observe from Figure 2 and Table 3, that a Bloom model trained on 1000 samples of CONLL improves on the CONLL task by over 30 points, but drops 1 point of general performance drop as measured by the Alpaca test set. To clarify this, we will extend l82 to include more details and add a clarification line in the introduction and examples in the appendix.
>
> The Task and Format Agnostism requirement (TFA) ensures minimal bias in the evaluation due to training data formatting artifacts (l93). In industrial scenarios, this is essential since many datasets that industrial actors could add to the training data (S_1), or fully train a model on (S_2) have specific response formatting that is not always fully similar to what a zero-shot model will answer. For instance, the label from a sentiment analysis dataset sample  (SST2) may simply state a movie review is “negative”, while a general-purpose model may output that the review “[...] appears rather positive from the adjectives used”. In this case, although the output is the same, most reference-based metrics will heavily penalize the extra details from the model output. To grasp model understanding capabilities of a target task independently of formatting artifacts, this requirement is crucial as can be seen in Figures 2, 3 and 4 which display experiments that simply cannot be run with metrics biased by output format. To clarify the need of TFA, we will detail the paragraph line 95 with examples, and add a clarification line in the introduction.
> In all industrial scenarios for IFT LLMs, rigorous model evaluation is necessarily linked to evaluation metrics that comply with both CAT and TFA which is one of the point this paper makes.
>
>
> ###  Clarifications on the industrial need to repurpose IFT models
>
> Repurposing an Instruction-following LLM for specific tasks  is common in the industry, either by keeping the LLM general-purpose but adding new capabilities (S1), or by fully specializing an LLM for a set of tasks (S2).  Examples of (S1) could be companies aiming to extend open-source LLMs with company specific needs (coding, internal FAQ answering capabilities) while retaining the benefits of a general purpose instruction models. This paper shows in Section 3.1 that this strategy is interesting because new capabilities can be introduced with minimal performance degradation on the other tasks. (S2) represents use cases in which a model is used for a specific purpose only (summarization for example). In this case, as shown experimentally in section 3.2, it is interesting to start off from an Instruction Following model rather than its base counterpart in low-data regimes where more performance can be extracted.
>
> Our results are especially exciting because we show repurposing IFT models is a strong industrial strategy, especially in terms of data efficiency, and uncover that the use of synthetic data can help optimize efficiency even more (Key Takeaways) .
>
> Hopefully we were able to clarify the reviewer’s concerns  and we will strengthen the paper by expanding on these points in the extra space available upon acceptance. We hope our explanations will convince the reviewer of the well foundedness of our work, and shed more light on our experimental choices and claims, hopefully augmenting their soundness and excitement scores.

---

### Meta-Review · Area_Chair_amhV · 2023-09-28

**Recommendation:** 4

**Metareview:**

The work introduced Instruction Fine-Tuning (IFT), a paradigm that enhances the zero-shot capabilities of Large Language Models (LLMs) by fine-tuning them on natural language instructions. It proposed two new requirements for metrics used to evaluate IFT models: Comparability Across Task (CAT) and Task and Format Agnostism (TFA). Showed that using LLMs as scoring agents is a good way to meet these requirements, and compares them with existing metrics such as BLEU, ROUGE, and GLUE. The work also demonstrated how synthetic data can be used to improve formatting skills, and how IFT models can reduce the need for expert data in industrial settings.

Pros:
1: Introduced new evaluation metrics that are more suitable for measuring the performance of IFT models across different tasks and formats, which can help researchers and practitioners compare and benchmark their models more effectively.
2: Provided practical insights and recommendations for fine-tuning IFT models on different tasks, and shows how synthetic data and IFT models can reduce the dependency on expert data in industrial settings, which can lower the cost and complexity of deploying IFT models in production.

---

### Decision · Program_Chairs · 2023-10-07

**Decision:**

Accept-Main

**Comment:**

The work introduced Instruction Fine-Tuning (IFT), a paradigm that enhances the zero-shot capabilities of Large Language Models (LLMs) by fine-tuning them on natural language instructions. It proposed two new requirements for metrics used to evaluate IFT models: Comparability Across Task (CAT) and Task and Format Agnostism (TFA). Showed that using LLMs as scoring agents is a good way to meet these requirements, and compares them with existing metrics such as BLEU, ROUGE, and GLUE. The work also demonstrated how synthetic data can be used to improve formatting skills, and how IFT models can reduce the need for expert data in industrial settings.

Pros:
1: Introduced new evaluation metrics that are more suitable for measuring the performance of IFT models across different tasks and formats, which can help researchers and practitioners compare and benchmark their models more effectively.
2: Provided practical insights and recommendations for fine-tuning IFT models on different tasks, and shows how synthetic data and IFT models can reduce the dependency on expert data in industrial settings, which can lower the cost and complexity of deploying IFT models in production.